# Generation of Bio-Based, Shape- and Temperature-Stable Three-Dimensional Nonwoven Structures Using Different Polyhydroxyalkanoates

**DOI:** 10.3390/polym17010051

**Published:** 2024-12-28

**Authors:** Tim Hiller, Frederik Gutbrod, Louisa Bonten, Marc Philip Vocht, Mehdi Azimian, Julia Resch, Christian Bonten, Malte Winnacker

**Affiliations:** 1German Institutes of Textile and Fiber Research (DITF), Koerschtalstr. 26, D-73770 Denkendorf, Germany; 2Institut für Kunststofftechnik, University of Stuttgart, Pfaffenwaldring 32, D-70569 Stuttgart, Germany

**Keywords:** sustainability, biopolymers, polyhydroxyalkanoates, poly(3–hydroxybutyrate) (P3HB), poly(3–hydroxybutyrate-co–3-hydroxyvalerate) (PHBV), poly(3–hydroxybutyrat–co–3–hydroxyhexanoat) (PHBH), melt processing, rheology, meltblow(n), nonwovens

## Abstract

Recent research has shown the potential of polyhydroxyalkanoates (PHAs), particularly poly(3–hydroxybutyrate) (P3HB), to form nonwoven structures with fine fiber diameter distributions ranging from 2.5 µm to 20 µm during the meltblow process. The shortcomings of existing fabrics of this type include high brittleness, low elongation at break (max. 3%), and a lack of flexibility. Furthermore, the high melt adhesion and the special crystallization kinetics of PHAs have commonly been regarded as constraints in filament and nonwoven processing so far. However, these two properties have now been used to elaborate a three-dimensional fiber arrangement on a matrix, resulting in the creation of dimensionally and temperature-stable “nonwoven-parts”. Moreover, this study investigated the PHA copolymer poly(3–hydroxybutyrate–co–3–hydroxyhexanoate) (PHBH), revealing a similar processability to P3HB and PHBV in the meltblow process. A significant increase in the (peak load) elongation in the machine direction was observed, reaching values between 5% and 10%, while the tensile strength retained unaltered. The addition of the bio-based plasticizer acetyltributylcitrate (ATBC) to PHBH resulted on an increase in elongation up to 15%. The three-dimensional fabric structure of PHBH exhibited complete resilience to compression, a property that differentiates it from both P3HB and PHBV. However, the addition of the plasticizer to P3HB did not lead to any improvements. This interesting array of properties results in moderate air permeability and hydrophobicity, leading to impermeability to water.

## 1. Introduction

The demand for bio-based and/or biodegradable alternatives to oil-based polymers is one of the most urgent topics of current polymer research. It should be noted that not all bio-based polymers (polymers created on the basis of biomass or monomers derived from biomass) are necessarily biodegradable (decomposing only into CO_2_ and water under certain conditions) and the other way around [1]. A promising class of polymers, combining both specifics, are polyhydroxyalkanoates (PHAs). Although over 100 different PHA variations are known today [2,3], poly(3-hydroxybutyrate) (P3HB) and some selected copolymers (mainly PHBV and recently PHBH) retain most of the industrial relevance due to their price and available scale of production. Meanwhile, a variety of biotechnological as well as synthetic methods are available for accessing these bio-polyesters [4].

PHA copolymers are frequently reported to exhibit superior properties to homopolymeric PHAs, primarily due to their ability to achieve lower crystallinities [5], which in turn results in reduced brittleness and stiffness [5,6,7,8]. Additionally, the use of copolymers enables more flexible processing due to their lower melt temperatures [5,9,10,11]. However, the use of copolymers does not entirely circumvent the issues associated with pure PHAs. These include the high costs, lower yields in synthesis [12,13], and the dependence of the property “improvement” on the co-monomer content in the structure [14,15].

The material properties of PHAs are still challenging, which is reasoned in their crystallite melt temperature being located close or slightly below the (thermal) degradation start temperature [16]. The description of the aging of PHAs in the form of chain scission and depolymerization due to thermal and/or mechanical influences represents a significant field of research, to which numerous publications have been dedicated [17,18,19] and which will only be referred to in this paper. Additional factors, such as high adhesion of the melt and low melt strength and stretchability, limit the usability of PHAs even further [20]. Moreover, the initial crystallization rate is coupled with a strong post-(secondary) crystallization process [21]. The latter results in high degrees of crystallinity (up to 80% [16]) with large spherulites [16,22], which consequently leads to a high brittleness [2,20], which is disadvantageous for industrial applications. In plastics technology, processing methods are therefore used to optimize the material properties. Typical processing tasks include the dispersion of additives and the filling or reinforcement of plastic melts. For this purpose, twin-screw extruders are mainly used. The mixing processes that occur here can be classified as dispersive and distributive mixing processes. Therefore, the screws have a modular design and are assembled in accordance with the specific requirements of the process [23].

Polylactic acid (PLA) is the most frequently reported and most industrialized member of the family of bio-based polymers. Although desirable meltblown fiber characteristics and nonwoven performance have already been achieved [24,25,26,27,28,29,30,31], its biodegradability is limited to industrial conditions (specific environmental conditions, e.g., high-temperature, etc.) [32,33,34,35] and is accompanied by the release of toxic residues [36,37]. The use of PHA-based nonwovens offers a number of advantages over PLA-based nonwovens. These include intrinsic hydrophobicity, a more comprehensive biocompatibility, faster biodegradability in flexible environmental conditions, greater flexibility regarding the applied co-monomers and copolymers, and better CO_2_ neutrality. In contrast, PLA-based nonwovens may have advantages over PHA-based nonwovens in terms of costs, market availability, and application-specific properties like higher flexibility and temperature resistance [9,38,39].

In previous research, we successfully demonstrated the meltblow processing of different P3HB grades and PHBV on a technical scale [40]. Previously, the meltblow processing of PHAs/P3HB was only reported on lab-scale equipment [41], with a limitation to very coarse fiber diameters (>10 µm) [42] or by use of huge amounts of plasticizers (5–15%), which resulted in maximum fabric elongations of 4% [41]. In our previous research, we achieved fiber diameters below 10 µm [40] with a median fiber diameter down to 2.4 µm. Additionally, the variability of the median diameter was demonstrated to range up to 20 µm through the adjustment of process settings, thereby illustrating the flexibility for potential application fields. The shortcomings of post-crystallization were addressed by developing targeted process settings and time-stable nonwoven fabrics (>1 year). While the fabrics’ flexibility was sufficient, the elongations at break remained a significant limitation, reaching a maximum of 3% in both the machine direction (MD) and the cross direction (CD). However, no improvement to fiber diameters and fabric properties could be achieved with PHBV, which may be attributed to the low co–valerate (3–hydroxyvalerate (3HV)) content of industrial-grade PHBV [40].

In this study, our objective was to investigate the potential of PHAs to form a stable three-dimensional (3D) fabric. This was based on the hypothesis that the combination of high melt adhesion and special crystallization kinetics could overcome the limitations commonly associated with PHAs in filament and nonwoven processing. These 3D structures were produced directly on the conveyor belt via the meltblow process and were designed to retain their stability over time and when subjected to certain temperatures. A plant-pot, as a simple yet application-relevant object shape, was chosen as a demonstrator. The produced pots were characterized and compared to purchasable reference materials.

Furthermore, this study examined two approaches to enhance the low elongation and flexibility of PHA nonwovens. Accordingly, the PHA copolymer poly(3–hydroxybutyrate–co–3–hydroxyhexanoate) (PHBH)—containing short-chain 3-hydroxybutyrate (3HB) units and medium-long-chain 3-hydroxyhexanoate (3HHx) units—was investigated regarding its usability for the meltblow process using a technical-scale meltblow line. Nonwoven webs were characterized for their base weight, thickness, average fiber diameter, air permeability, and their mechanical performance (tensile test). Furthermore, the second approach to improve the properties involved the addition of a bio-based plasticizer (acetyltributylcitrate (ATBC)) and a bio-based chain-extender (epoxidized linseed oil (ELO)) to P3HB and PHBH. The compounds were produced using a twin-screw extruder and the materials were characterized based on their rheological and thermal behavior.

## 2. Materials and Methods

### 2.1. Materials

#### 2.1.1. Commercial Polymers

P3HB “P316” was obtained from Biomer Biopolyesters (Schwalbach, Germany). Its melt flow index (MFI) is specified as 10 g 10 min^−1^ at 170 °C, 2.16 kg (ISO 1133) [40,43]. This material has a broad molar mass distribution and was comprehensively characterized in an earlier publication [40].

The chemical structure of P3HB is displayed in Figure 1.

PHBH (chemical structure, see Figure 2) “BP350–15” from BluePHA^®^PHA (Beijing, China) was purchased from Helian Polymers BV (DK Belfeld, The Netherlands). According to the manufacturer’s specifications, at a 3-hydroxyhexanoate (3HHx) content of 10%, this polymer exhibits a melt flow index of 10–15 g 10 min^−1^ at 165 °C, 5.0 kg (ISO 1133), a melting temperature of 133 °C, a density of 1.19 g cm^−3^, and a glass transition temperature of −2 °C [44].

#### 2.1.2. Additives

Two different fully bio-based additives were used in order to optimize the material properties. The aim of both was to reduce the viscosity to a range that enabled stable processing in the meltblow process and did not compromise the bio-character. Acetyltributylcitrate (ATBC) (Jungbunzlauer Ladenburg GmbH, Ladenburg, Germany) as well as an epoxidized linseed oil (MERGINAT ELO) (HOBUM Oleochemicals GmbH, Hamburg, Germany) were selected. In addition to its function as a plasticizer, the epoxidized oil provided the potential to react with the hydroxyl and carboxyl groups of the PHAs in order to counteract their degradation process.

#### 2.1.3. Reference Nonwoven Fabrics

Meltblown nonwoven fabrics of P3HB (Biomer “P316”) and PHBV (“Y1000P”, TianAn Biologic Materials Co., Ltd., Ningbo, China) from an earlier study [41] were taken as reference materials. The fabrics were processed using the same machine and setting as for the polymers in this study. The samples are labeled as “P3HB-Ref.” and “PHBV-Ref.” in the following.

#### 2.1.4. Commercial Plant Seed Pots (Demonstrator References)

Plant seed pots, made from natural fibers, were purchased commercially to provide a direct comparison of the targeted demonstrator application to market-available products which are also of a bio-based and biodegradable nature. Therefore, cellulosic (wooden-based) plant seed pots “Anzuchttöpfe rund 6 cm Ø” and coconut fiber-based pots “Proflora Kokosfaser Anzucht-Pflanztopf 0,30 L” of the same dimensions were obtained from OBI Home and Garden GmbH (Wermelskirchen, Germany).

### 2.2. Compounding Experiments

A ZSK 26 twin-screw extruder (Coperion GmbH, Stuttgart, Germany) with co-rotating screws was used for the compounding experiments. A proportion of 10 wt.% for the additives was determined on the basis of preliminary tests. Various screw configurations with different proportions of mixing elements and temperature profiles were tested in order to achieve gentle processing with good homogenization. The final screw configuration and temperature profile are shown in Figure 3. In addition to the typical conveying elements, this screw configuration consisted of distributive and disruptive mixing elements as well as a back-pressure element.

Further processing parameters included a dosing rate of 4 k h^−1^ for the polymer and a rotational speed of the twin-screws of 120 rpm. After compounding, the compounds were dried for at least 24 h at 40 °C in a vacuum chamber.

### 2.3. Meltblow Process

Nonwoven processing trials were conducted on a technical-scale line with a working width of 500 mm. The line consisted of a single-screw extruder (3 zone screw, ∅ 20 mm × 20 D) from Extrudex GmbH (Mühlacker, Germany) and a gear pump from Mahr Metering Systems GmbH (Göttingen, Germany) with a volume of 0.6 cm^3^ rpm^−1^ to melt and transport the polymer to the spinning beam with a maximum throughput of 4 kg h^−1^. The air system comprised a compressor (Aertronic D12H) from Aerzener Maschinenfabrik GmbH (Aerzen, Germany) with an air volume flow limit of 220 Nm^3^ h^−1^ (minimum) and 325 Nm^3^ h^−1^ (maximum) in conjunction with a flow heating system produced by Schniewindt GmbH & Co. KG (Neuenrade, Germany). The spinneret was a 561-hole Exxon-type die with a width of 500 mm (28.4 holes per inch (hpi)) and nozzles 0.3 mm in diameter (L/D = 8). The maximal die pressure of the spinneret was set to 50 bar with a safety limit of 45 bar. The set-back between the nozzle tip and the air blades was 1.2 mm and the end gap was set to 2.0 mm for all trials. The conveyor belt manufactured by Siebfabrik Arthur Maurer GmbH & Co. KG (Mühlberg, Germany) was a steel fabric tape in canvas weave with a clip seam, measuring a total width of 0.72 m (no. 16 cm^−1^ linen weave) with a stainless steel (1.4404 AISI 316L) warp and weft wire 0.22 mm in diameter. The maximum take-up velocity was 10 m min^−1^ and the height relative to the die could be adjusted from 200 mm up to 500 mm to vary the die–collector distance (DCD). Below the belt section, where the filaments were laid down, an air-suction box (suction surface of 0.128 m^2^, 0.20 m × 0.64 m) with a maximal suction volume of 2900 Nm^3^ h^−1^ (maximum flow velocity: 11 m s^−1^) was placed to remove the process (and secondary) air and to support the web formation on the belt.

Meltblown nonwovens were produced with the polymers listed in Section 2.1 and Section 2.2, varying the process temperatures and polymer throughput as the main parameters to reveal a stable process window. The melt temperature was adjusted over the temperature of the die and the spinning head based on the results of the rheological characterization (see Section 2.6) of the respective material with the objective of achieving a zero-shear viscosity at a process temperature of less than 100 Pa s. Furthermore, adjustments were performed during the experiments in order to obtain constant fiber formation at the die and a homogeneous shot-free laydown on the conveyor belt.

The process air throughput was varied between the minimum and maximum output (220–325 Nm^3^ h^−1^) of the compressor in order to define the possible diameter range for each polymer at the respective process setting. Due to the delayed and slow start of the crystallization of PHAs, the process air temperature was maintained 5 K below the melt temperature and the DCD was kept constant at a maximum of 500 mm. This approach was taken to avoid sticking or depletion of the deposit on the conveyor and to reduce fiber-to-fiber bonding [40]. The collector speed was adjusted in accordance with the polymer throughput in order to produce a constant area and base weight of the produced nonwovens, with a target of 100 g m^−2^. This was intended to ensure the comparability (without influence of the base weight) of web properties under different process settings.

In summary, the following parameters of the entire system were used as variables for this study:Polymer throughput: max. 3.5 kg h^−1^;Process temperature (polymer, air): potentially up to 420 °C;Air throughput: max. 325 Nm^3^ h^−1^;Collector speed: max. 10 m min^−1^

### 2.4. Producution of Three-Dimensional Meltblown Samples

Three-dimensional structures were produced using a “counter-shape”, which was placed on the conveyor belt prior to the deposition point of the polymer stream. A plant pot 55 mm (opening) in diameter (50 mm in height; see Figure 4a) was chosen as an application-oriented demonstrator shape, based on commercial biodegradable cellulosic reference plant pots (diameter 6.0 cm, height 65 mm; see Figure 4b).

After passing through the stream, the pot was rotated by 90° and placed on the start point again. In total, four passes were carried out with the belt speed adjusted to a base weight of 35 g m^−2^ in order to obtain a total base weight of 150 g m^−2^. After the last passage, the parts were taken from the conveyor belt and the nonwoven removed from the counterpart (after around 1 min). The process is schematically shown in Figure 5.

### 2.5. Material Drying and Determination of the Moisture Content

Prior to testing or processing, all materials were subjected to pre-drying in an oven at 80 °C for at least 6 h under fine vacuum condition (<1.8 10^−1^ mbar).

The residual water content for all polymers was determined by means of Karl Fischer titration, which was performed on an “899 Coulometer” and an “885 Compact Oven SC” (both manufactured by Deutsche METROHM GmbH & Co. KG, Filderstadt, Germany) at 140 °C. The resulting water content was required to be <150 ppm, which was achieved for all types of polymers/compounds.

### 2.6. Polymer Characterization

Shear rheological experiments in time-sweep modes were performed on a Discovery HR-2 rheometer (TA Instruments, New Castle, DE, USA) using a plate–plate geometry at a temperature of 180 °C. The material was placed on the lower plate (25 mm in diameter) and the gap was adjusted to 1.0 mm. Subsequently, the excess material was removed, and the test was carried out under a nitrogen atmosphere (50 mL min^−1^) with 5% elongation and an angular frequency of 1 rad s^−1^ over a period of 10 min.

Additionally, the thermal properties were determined using different methods. A DSC 2/400 (Mettler-Toledo GmbH, Gießen, Germany) was employed for the DSC analysis, whereby the samples were analyzed in a temperature range from −80 °C to 200 °C at a heating and cooling rate of 10 K min^−1^. Two heating phases and one cooling phase were carried out, whereby an isothermal phase at 200 °C was omitted in order to reduce the influence of thermal damage on the material. A TGA 3+ (Mettler-Toledo GmbH, Gießen, Germany) was used for the thermogravimetric analysis. The samples were analyzed in a temperature range from 30 °C to 900 °C at 10 K min^−1^ under normal air atmosphere.

### 2.7. Nonwoven Testing

#### 2.7.1. Fabric Area Base Weight

The area base weight was determined referring to DIN EN ISO29073-1, adjusted by cutting out and weighing 100 cm^2^ square sections (10 cm × 10 cm). To consider homogeneity scattering in the cross direction (CD) of the nonwovens, three samples with the dimensions 10 cm × 10 cm were taken in the CD and averaged.

#### 2.7.2. Nonwoven Thickness

The fabric thickness was measured in the samples used for the base weight measurements using a test head (Frank-PTI GmbH, Birkenau, Germany) of 25 cm^2^ and a test force of 5 cN cm^−2^. Eight measurements were conducted diagonally along the sample, determining a median value for thickness (*δ*).

#### 2.7.3. Air Permeability

In accordance with the base weight sampling, the air permeability was measured on the 10 cm × 10 cm sections in accordance with EN ISO 9237:1995-12 with a sample size of 20 cm^2^ and a differential pressure of 200 Pa.

#### 2.7.4. Scanning Electron Microscopy/Fiber Diameter (Distribution)

The fiber diameter distribution was determined by means of scanning electron microscopy (SEM). Therefore, a circular sample was punched out of the nonwoven and placed on the SEM carrier, sputtered in argon plasma (40 s under a vacuum of 0.1 mbar, with a distance of 35 mm, a current of 33 mA, and a voltage of 280 V) with a gold–palladium layer of 10–15 nm. Three SEM micrographs per sample were taken with a magnification of ×500, using a “TM-1000 tabletop electron microscope” of Hitachi High-Tech Corporation (Tokyo, Japan). The accelerating voltage was 15 kV in the “charge-up reduction mode”. The magnification was selected to enable the capture of around 40 individual fibers per image. Contrast and brightness were adjusted to obtain an image of straight monochromic fibers in front of a dark monochrome background. To analyze the images with regard to automated fiber diameter distribution, the beta-software “MAVIfiber2d, v.1.1” of Fraunhofer ITWM (Kaiserslautern, Germany) was used [46]. Initially, the images were smoothed by an algorithm and binarized by the software, after which a statistical analysis was conducted on each fiber pixel without segmentation into individual fibers [47,48]. After merging the output of the three images, the mean and median fiber diameter as well as the standard deviation and interquartile range were determined.

#### 2.7.5. Mechanical Properties

Tensile tests of the nonwovens were carried out on an “Instron UPM 4301” of Instron GmbH (Darmstadt, Germany) to determine the tensile strength (*σ_m_*) and the elongation at peak force (*ε_m_*) and at break (*ε_B_*) of the nonwoven fabrics in the MD (machine direction) and CD (cross direction), as well as the Young’s modulus (E) as the secant modulus. For each sample, five specimens with a width of 15 mm were cut out in the MD and CD and tested. The 3D demonstrators (pots) were sliced open along one edge to form a flat “sheet” and five specimens were cut out randomly.

The sample thickness was determined individually in accordance with the specifications set forth in DIN EN ISO 9073-2 and the median of five measurements was used for the calculation of the stress from the recorded force. The tests were conducted with 100 mm min^−1^ using a 5 kN measuring head with pneumatic clamps (clamping length of 100 mm). The tenacity was calculated for the sample dimensions, the fabric thickness, and the measured peak force. The median and the standard deviation of all measured properties were employed to facilitate a comparative analysis of the nonwoven characteristics.

#### 2.7.6. X-Ray Diffraction

Wide-angle X-ray diffraction (WAXD) measurements were recorded on a “D/Max Rapid II diffractometer” (Rigaku Corp, Akishima, Japan), equipped with a 0.8 collimator and an image plate detector using monochromatic Cu *Kα* radiation (*λ* = 0.15406 nm; *U_acc_* = 40 V; *I_acc_* = 30 mA). A scanning rate of 0.2° min^−1^ and a step size of 0.1° were applied. The measurement time was one hour for all samples under investigation. Background correction was performed using a blank measurement and the resulting scatter images were converted into the corresponding diffractograms using 2*θ* intensity conversion. The diffraction patterns were analyzed using the PDXL 2 software (version 2.3), and pseudo-Voigt profile fitting was chosen for the evaluation of reflex positions and crystalline fraction determination. The degree of crystallinity *χ_c_* was calculated according to Equation (1),
(1)$$\chi_{c}=\frac{\sum I_{c}}{\sum \left(I_{c}+I_{a}\right)}$$
where *I_c_* and *I_a_* are the integrated intensities of crystalline reflexes and amorphous reflexes, respectively.

The samples were prepared by arranging nonwoven sheets parallel to the carrier.

#### 2.7.7. Thermal Stability

To determine the time and temperature stability, nonwoven fabrics (flat sheets) were tested in relation to their heat shrinkage according to the “drying oven method” (ISO 11501:1995 and GB/T 12027-2004) [49]. Rectangular samples (300 mm × 50 mm in MD) were punched out on the left, in the middle, and on the right side of a nonwoven sample. The specimens were placed free-hanging in an oven at 120 °C. After reaching the specified time of 15 min, the samples were taken out and the size of the samples was measured. The ratio of the dimensional change value to the size before shrinkage was calculated as the percentual shrinkage rate of the sample.

Furthermore, three-dimensional nonwoven demonstrators were placed into an oven with a defined temperature, starting with 100 °C. The parts were kept in the oven for 15 min and the dimensions were measured before and after “tempering”. When no changes were observed, the procedure was repeated at a 10 K higher temperature.

#### 2.7.8. Qualitative Evaluation of the Flexibility and Shape Retention of 3D Nonwoven Structures

To evaluate the flexibility of the three-dimensional nonwovens qualitatively, the pots were loaded with a weight of 400 g. After a loading time of one minute, the weight was removed and the shape retention was quantified based on the ratio of the initial to the final pot height (see illustration in Figure 6).

Thereby, the original form was labeled as *l*_0_. During loading, the shape was compressed (*l*_1_) due to the weight force (*F*). When the weight was removed from the shape (*l_e_*), the relieved shape was compared with the previous initial shape. The differences in height were measured in the z-coordinate.

## 3. Results and Discussion

### 3.1. Characterization of the Compounds

Figure 7 illustrates the DSC measurements through the use of additives. Multiple peaks, as they appear in the thermograms of P3HB and PHBH, have already been identified in other studies. These may be attributed to a combination of melting, recrystallization during heating, different crystal modifications (polymorphism), different molecular weights, different morphologies, and the physical aging or relaxation of the rigid amorphous part. [50]. A broad and blurred melting range with two peaks was identified for P3HB. According to earlier findings [40], this phenomenon can be attributed to a broad molecular weight distribution, resulting in an inhomogeneous melting behavior.

It can be seen that the melting temperature and crystallization temperature decrease by using both additives. While endothermic peaks at 165.7 °C and 158.2 °C were identified for “pure” P3HB (**a**), these were reduced by around 1–2 K when using 10 wt.% ELO (**b**) and by around 3–5 K when using 10 wt.% ATBC (**c**). This phenomenon can be explained by the improved mobility of the polymer chains due to the additives, whereby the secondary valence forces are reduced allowing for melting at lower temperatures. In comparison, the exothermic peak of 107.2 °C of “pure” P3HB was reduced by 3 K (ATBC) and by 1 K (ELO). The incorporation of additives impedes the formation of tight molecular aggregates, thereby postponing the onset of crystallization. Acetyltributylcitrate exhibits a markedly superior plasticizing effect compared to the epoxidized oil extending the processing window to lower temperatures. For this reason, only acetyltributylcitrate (ATBC) was considered for PHBH.

For PHBH, enthalpy relaxation after the *T_g_* and several exothermic effects were observed (Figure 8). This leads to the assumption of slow and incomplete crystallization after the first heating process, with post-crystallization processes and crystal perfection taking place in the subsequent melting process.

Despite the complex behavior, significantly earlier melting was observed compared to P3HB. As PHBH is a copolymer comprising 3HB and 3HHx units, this can be attributed to the steric hindrance caused by the propyl group. Depending on the proportion in the copolymer, this group hinders the dense packing of the polymer chains, which reduces the secondary valence forces and promotes early melting. Since, according to Eraslan et al. [51] and Volova et al. [52], a PHH homopolymer is an amorphous material, the melting range can be attributed to the crystallites of the 3HB component. With regard to the chain degradation of PHAs due to increased temperatures, the reduced melting temperature allows for processing at lower temperatures. This can counteract chain degradation at elevated temperatures.

Table 1 presents the results of the TGA measurements (onset temperature *T_on_*, the temperature at 10% mass loss *T*_10%_, and the endset temperature *T_end_*) of the degradation stage of the various materials. The decomposition of P3HB takes place over a broader temperature range, which is why a higher temperature with a mass loss of 10% and a higher endset temperature could be identified despite a lower onset temperature. In addition to a broad molecular weight, which could already be observed in [40], this result may also be attributed to the presence of additives that only decompose at elevated temperatures. The use of the citrate ATBC generally resulted in increased degradation at lower temperatures. The degradation stage is characterized by a “smearing” effect, which can be attributed to the mixture of the polymers with the citrate, resulting in a multimodal and broad molecular weight distribution.

The use of an epoxidized oil did not result in less degradation overall compared to the untreated material. Given the assumption that the epoxy group is capable of reacting with the respective chain ends, the usage of an epoxidized oil should delay the progressive depolymerization. Only T_end_ showed a higher temperature, which could be explained by the higher decomposition temperature of the epoxidized oil itself. It was previously found by Park et al. [53,54] that the carboxylic acid groups from the decomposition of PHAs can react with epoxide groups to form networks between the chains and retard degradation.

According to state of the literature, reviewed by Eraslan et al. in 2022 [51], PHBH is said to have increased thermal stability due to its 3HHx content. For PHBH with a 3HHx content of 10 mol.%, the TGA consistently showed the lowest temperatures for the course of decomposition. This is particularly evident in the onset and endset temperatures. Thus, contrary to assumptions, the lowest thermal stability was identified for PHBH.

Figure 9 shows the complex viscosities of the different materials over time. The observations from the DSC measurements are also represented in the rheological analysis. As can be seen, the use of external plasticizers in commercially available PHAs enables melting at lower temperatures and thus reduces viscosity. Therefore, the impact of plasticizers is evident in the vertical shift in the curve towards lower viscosities by allowing the chains to slide away from each other more easily. This development is significant for PHAs, as these react sensitively to elevated temperatures, especially when accompanied by mechanical stress. A reduction in the initial viscosity of approx. 59–64% was achieved with the various materials using ATBC.

In contrast, the epoxidized linseed oil only has a minor influence here too. In addition to the molecular structure, the molar mass can also be cited as the reason for the different behavior of the additives. As acetyltributylcitrate has a significantly lower viscosity than the epoxidized oil, the viscosity is reduced to a greater extent by the addition of the same proportion, irrespective of the softening effect. A lower level of degradation due to reactions with the epoxy groups could not be observed.

In addition, the degradation process of the polymers in the rheological analysis also leads to a further drop in viscosity over time. This is particularly pronounced in the case of PHBH, while the viscosity of P3HB decreases to a lesser extent. It is well known that the rheological behavior not only depends on test parameters such as temperature and the constitution of the polymer. With an increasingly broad molecular weight distribution, the low-molecular-weight components also act as a lubricant and promote the flow of the plastic melt [55]. Since, according to [40], P3HB already has a broad molecular weight distribution, this could explain the already low initial viscosity. However, since PHBH has a melting range at lower temperatures than P3HB, a lower viscosity was also anticipated at the set testing conditions. However, factors such as the molar mass itself also play an important role, which is why a higher molar mass can be assumed for PHBH.

### 3.2. Investigation of the Meltblown Processability

Based on the observed differences in the material properties of the three PHA types (P3HB, PHBV, and PHBH) and the PHA compounds (compare Section 3.1 and [40]), differences in their processing behavior and/or the requirement of adaption in the processing parameters (required temperature, limitation in throughput, and accessible fiber diameters) can be assumed. This was already highlighted by the melt flow (rheological) curves and the relevant characteristics of the materials at 180 °C compared in Table 2.

While the melt flow index (MFI) is a weak parameter [40,57], the viscosity in the Newtonian (shear-independent) regime serves as an established tool for the prediction of the processability. However, as PHAs are subject to a strong thermal degradation at temperatures close to their melting temperature, the reliability of frequency and temperature sweeps is negated. For this purpose, the start values (*η_to_,* G′*_to_,* G″*_to_*) as unaffected measured values were taken from tests in the time-sweep mode under low-shears condition. These were supplemented by the viscosity value corresponding to a characteristic residence time in the extrusion process (~300 s) *η*_300_*_s_*. The final resulting processing temperatures of each material and the respective rheological properties are presented and compared in Table 3.

While the virgin PHA types show higher start viscosities above the processing window for meltblow processing (~<150 Pa s), which one can achieve within a typical extrusion time, the processed compounds “P3HB+ELO”, “P3HB+ATBC”, and “PHBH+ATBC” already lie below the upper processing limit and thus are in the processable window. Further, it can be noted for PHB that the plasticizing additives cause a lower process temperature due to the reduction in viscosity. However, PHBH, supplemented with ATBC, also shows a significant reduction in viscosity, but no reduction in the processing temperature was possible due to a high process pressure level, although the viscosity dropped significantly over the 300 s residence time.

All processable settings (at the respective lowest possible process temperatures enabling a homogeneous fiber deposition) are given in Table 4.

As for the virgin polymers, the P3HB compounds allowed a higher flexibility of the throughput, showing the possibility to achieve higher per-hole throughput levels >0.1 g min^−1^. Unmodified PHBH was limited to low or moderate throughputs at a quite high process temperature of 190 °C, although its melting temperature was intrinsically lower. For this, a too-high mean polymer chain length can be assumed in correlation with the significant higher melt viscosity at 180 °C compared to P3HB and even PHBV, as characterized in a previous work [40]. However, as the processing temperature of PHBH was almost as high as for PHBV, the use of PHBH and the compound PHBH + ATBC resulted in a more stable meltblow process and a more homogeneous fiber deposition due to the higher difference between its processing temperature and its melting temperature.

### 3.3. Nonwoven Characteristics

Figure 10 shows the SEM micrographs at 100× magnification for the reference nonwovens from [40].

A too-high difference between the air temperature and the temperature of the melt (10 K vs. 5 K) for P3HB–01 (Figure 10b) led to a too-coarse fiber deposition, while a too-high air amount led to a highly nonuniform fiber diameter distribution of fine and coarse fibers (PHB–03; Figure 10c) due to higher air flow turbulences in the process. This was also the case for PHBV (Figure 10d), but at a higher level due to the harsher processing temperatures.

Figure 11 shows the respective SEM micrographs for all produced nonwoven samples from virgin PHBH and the P3HB and PHBH compounds.

PHB, compounded with epoxidized linseed oil (ELO), resulted in a deliquescing of the deposited fibers and “thermal branching” (merging of fibers due to flow reasoned by insufficient cooling [27,41]) at a lower throughput of 0.039 g ho^−1^ min^−1^ (P3HB+ELO–01 and P3HB+ELO–02; Figure 11a,b). At a higher throughput of 0.077 g ho^−1^ min^−1^, the fiber deposition was more homogeneous and the average fiber diameters were similar to those of P3HB from previous works [40,58]. Therefore, a higher difference between process air temperature and melt temperature (12 K vs. 5 K, compare Table 4) successfully achieved a finer and denser fiber deposition (P3HB+ATBC–03, Figure 11k).

The test results of the characterization of the nonwovens are given in Table 5.

Comparing the P3HB compounds to the virgin P3HB showed a similar mechanical performance. A slightly higher tenacity could be achieved with the setting P3HB+ELO–02. However, the elongation of fabrics could not be increased by compounding P3HB with the bio-based plasticizers. Thus the main intended purpose of their use was not successful at this point. Nevertheless, for the compounds of P3HB with ELO, an increase in the Young’s modulus (~10%) could be observed when less polymer throughput was applied. This is due to the drag force of the process air acting with higher efficiency on a lower amount of melt (due to higher polymer chain orientation) until solidification. The air permeability was lower for all P3HB compounds, which also correlates with a higher resulting base weight, which derivates by up to two times (>200 g m^–2^) from the set target value due to a narrower fiber deposition on the conveyor belt (<500 mm vs. 550 mm). A potential reason for this is the formation of a denser air curtain around the nominal lower viscous melt.

The fiber diameter range, quantified by the mean and the median of the fiber diameter distribution, is similar to that of the samples presented in previous works [40,41,42]. The average fiber diameters of the PHBH fabrics, as well as the fabrics of the modified polymers, are slightly higher. However, the sensitivity of the fiber diameter to process parameter changes appears to be reduced. The fiber diameter variation displayed by the samples (ratio of the median to the mean average) is also in a standard range for meltblown nonwovens. In this context, the meltblowing process generally generates broader fiber distributions than other processes, e.g., the melt spinning of yarns. This is due to the complex interactions between the polymer melt and the turbulent air flow [59,60].

Replacing P3HB with PHBH results in a higher tenacity at comparable fiber diameter averages and air permeability values. However, the flexibility of fabrics improved drastically (qualitatively), and the elongation at maximum force could be successfully raised to 5% in the machine direction (MD). The fabrics of the PHBH compounds showed a lower tenacity and became denser, while also showing reduced air permeability, but the elongation further increased to >10%, thus fulfilling this study’s hypothesis.

Summarizing the characterization of the nonwovens, the key nonwoven characteristics (tenacity, elongation, modulus, and base-weight-standardized air permeability), selected for the best values for each material, are compared in Figure 12 for the different materials used.

As pointed out before, nonwovens made from virgin PHBH showed the highest tenacity (in both MD and CD), while PHBV was clearly limited in its mechanical strength (Figure 12a). PHBV also showed the lowest elongation, while this parameter was significantly increased for PHBH and further so for the compound of PHBH with ATBC (Figure 12b shows the elongation at break in contradiction to Table 5 at max. force). The Young’s modulus (Figure 12c) was again low for PHBV, but superior for P3HB when combined with ELO. The air permeability, standardized to the base weight to ensure comparability (Figure 12d), was significantly higher for both reference materials, virgin P3HB and PHBV, compared to the materials presented in this study, which may be significant for the choice of application. However, virgin PHBH still has a higher ratio of air permeability than the compounds used.

Figure 13 shows the results of the wide-angle X-ray diffraction, performed on a selected sample of each material, in order to obtain information about the crystallization status (degree of crystallinity and crystal structure) of the fabrics.

As already obtained in our previous work [41] (see the samples “PHB–Ref” and “PHBV–Ref” in Figure 8b), all X-ray diffraction images showed a semi-crystalline pattern. Several crystalline reflections occurred at 2*θ* ≈ 13.5° ((020)), 16.9° ((110)), 19.7° ((101)), 21.5° ((111)), 25.3° ((130)), 27.0° ((040)) and 30.0° ((002)) in accordance with the literature [3,40,61].

Differences in the process settings (throughput rates, air amounts, etc.) did not play a role in the crystallization, which is characteristically delayed for PHAs. However, the physics of the meltblow process with high forces from the air stream resulted in high orientation, leading to highly crystallized structures. The crystallinity of all samples was determined by fitting (pseudo-Voigt profiles) the diffractograms in the range of 2*θ* = 10–40°. The crystallinity for the P3HB samples was between 71% and 77%, that of OHBV was between 75% and 80%, and that of the PHBH samples was between 60% and 77%. The plasticizers showed limited influence on the peak positions and crystallinity.

In comparison to [3,40], the identified peaks indicate the predominance of orthorhombic α-crystals, with hexagonal β-form crystals (shoulder at *2θ* ≈ 20°, (101)-lattice) also present. This shoulder is indeed more pronounced for P3HB/PHBV and very diffuse for PHBH. Referring to Hufenus et al. [62], β-crystals develop stress introduced in the amorphous P3HB regions between lamellar α-crystals.

The presence of the shoulder at 2*θ* ≈ 20° ((101)-lattice) is typical for introduced chain orientation, such as uniaxial stretching [3,63], stress-induced crystallization [10,64], and the distribution of different crystal sizes. Another reason could be random chain scission during degradation [17,64,65], which is very common in the processing of PHAs, forming crystals with different lamellar thicknesses and thus different melting kinetics. However, the results indicate that the copolymer prefers to form α-crystals rather than β-crystals.

Furthermore, the materials were tested regarding their heat shrinkage. Due to the melting temperature of PHAs, the standard testing temperature (200 °C) was modified to 120 °C. The results are given in Table 6.

The heat shrinkage of all materials lies in the same range. Furthermore, the dimensional change is at such a low level that heat shrinkage can (almost) be excluded for all PHA types and compounds tested. For PHBH, this behavior of the nonwoven material contradicts the thermal behavior of the granules (see Section 3.1, Figure 9), which already showed two melting peaks of lower enthalpy at 75 °C (*T_m_*_1_) and 110 °C (*T_m_*_2_), before the final melting peak at 130 °C occurred. These peaks could be reproduced by measuring the nonwoven material, but with the absence of a previous recrystallization (see Appendix B). This indicates that the crystal structure corresponding to (*T_m_*_3_) is dominant after meltblow processing due to the high orientation. Furthermore, the enthalpy content of the peak corresponding to T_m3_ (~18 J g^−1^) is significantly higher towards *T_m_*_2_ (2.0 J g^−1^) and negligible for *T_m_*_1_ (0.3 J g^−1^).

### 3.4. Processing of Three-Dimensional Nonwoven Structures

The principle of three-dimensional meltblown structures was previously demonstrated by Farer et al. in 2003 [66]. However, they used a small lab-scale die with a width of 78 mm and a six-axis robotic arm and an additional external axis to spray fibers on a 3D-carrier structure/layer. This requires facilitating an independent machine set-up and is not applicable to standard (industrial) meltblow lines.

The high melt adhesion and the delayed crystallization, two parameters often cited as major drawbacks of PHAs, helped to successfully deposit the nonwoven structure on a three-dimensional counter-piece and to remove the resulting 3D-nonwoven structure from it without damage and without loss of the applied shape. Polymers showing a strong initial crystallization, such as polypropylene, do not show this described behavior, as the finest fibers generated in the meltblow process have already cooled too much before they hit the counter piece and lose the applied shape after being withdrawn.

In Table 7, the characteristics of plant seed pots made from P3HB, PHBV, and PHBH are compared to those of two commercial reference pots (of the same dimensions) made from natural fibers (cellulose and coconut fibers).

Comparing the mechanical properties, the tensile strength of the cellulosic pots could be achieved with P3HB and PHBH and the lower strength of the coconut pots could also be achieved with PHBV. The elongation is low for all materials, including the references, and superior for PHBH. Moreover, the Young’s modulus was the highest for PHBH and comparable for P3HB and cellulose fibers, as well as for PHBV and coconut fibers. It is noteworthy that the base weight of the PHA pots is only half that of the cellulosic and around 33% that of the coconut fiber pots, combined with their significantly lower wall thicknesses, while offering at least equal mechanical performance. PHBH has the lowest thickness due to its denser structure. While the air permeability is, as before, high for P3HB, the reference pots are almost impermeable for air. Filling the plant pots with water reveals another interesting feature of the PHAs. Due to their intrinsic hydrophobicity, the water remains in the pots for at least 50 h before the pots become water-permeable.

As with the flat sheet nonwovens, the three-dimensional nonwoven structures were tested with regard to their thermal stability. As all the materials used showed equal results in the heat shrinkage test previously, only P3HB was tested in this test. Plant pot demonstrators were exposed to different temperatures starting at 100 °C for 15 min in an oven. After removing these samples from the oven, they were examined for dimensional changes. Photographs of the 3D-nonwovens after exposure to different temperatures are presented in Figure 14.

Starting at 100 °C (Figure 14a) up to 130 °C (Figure 14e), the plant pots showed no change in their shape or their dimensions. The plant pot exposed to 140 °C (Figure 14f) showed loss of its shape after five minutes of exposure and the pot that had been exposed to 150 °C (Figure 14g) showed significant signs of decomposition, while the samples placed in the oven at 160 °C decarbonized entirely without residues (residues sucked away by the suction of the oven or spreading of the melt on the firebrick substrate) after 1 min of exposure to these conditions. As the meltblown fibers were identified to be highly crystallized, this behavior is not based on usual shrinkage mechanisms, but on the onset temperature of the melting region, which lies between 125 °C and 140 °C (see Section 3.1, Figure 7 and Figure 8). The higher surface to volume ratio of fine meltblown fibers can explain their higher affinity for faster melting at lower temperatures compared to the granular material.

In addition to the thermal stability, the structural stability and flexibility of the three-dimensional structures were also tested. For this purpose, plant pots made from P3HB, PHBV, and PHBH were compressed with a load of 400 g and their ability to unfold after compression was quantified as dimensional differences from the original state. The qualitative results are given in Table 8.

In terms of mechanical characteristics, the PHBH pots show a superior behavior to P3HB (and PHBV). The structure straightens itself out completely after unloading, whereas the P3HB and PHBV pots remain compressed or break easily under compression load. Thus, PHBH reveals a superior flexibility (mechanical resistance) as well as significantly higher ductility (elongation). This flexibility is not only based on elasticity, as the modulus of the PHBH fibers is in the same range as other (brittle) materials, but may be based on the fiber-to-fiber interactions and the fiber network formed in the meltblown deposition. Chemically, this may also be due to the longer alkyl-side chains of the hexanoate building blocks in the polymer chain. These are the reasons for PHBH’s lower crystallization [67] and correspondingly the higher proportion of amorphous regions in PHBH, which introduce more flexibility and ductility. Additionally, the higher chain mobility of the copolymer PHBH compared to the homopolymer P3HB, displayed by the lower crystallite melting temperature and especially the lower *T_g_*, contributes to its higher flexibility and ductility at room temperature.

## 4. Conclusions

In this study, the great potential of polyhydroxyalkanoates (PHAs), especially poly(3–hydroxybutyrate) (P3HB) and poly(3–hydroxybutyrate–co–3–hydroxyhexanoate) (PHBH), to form different nonwoven structures with fine fiber diameter distributions in the meltblow process was demonstrated. The DSC characterizations showed that both P3HB and PHBH exhibit complex thermal and crystalline properties. PHBH generally shows an earlier melting behavior compared to P3HB. The use of additives like epoxidized linseed oil (ELO) and acetyltributylcitrate (ATBC) influenced the thermal behavior by lowering the melting and crystallization temperatures. ATBC proved to be a more effective plasticizer than ELO and extends the processing window to lower processing temperatures. The complex viscosity analysis proved that the results of the thermal characterization are reflected in the rheological properties. Overall, the lower melting temperatures allowed for gentler processing, minimizing thermal chain degradation.

Both polymers were able to produce dimensionally stable three-dimensional nonwoven shapes in a one-step meltblow process, which also showed thermal stability up to 130 °C. Plant seed pots made from P3HB that were produced as a demonstrator application showed equivalent mechanical strength (and elongation) to commercial natural-fiber-based pots, as well as a lower area base weight and wall thickness. Furthermore, another interesting property of PHA nonwoven structures results in moderate air permeability and hydrophobicity, leading to impermeability to water.

Processed P3HB compounds showed greater flexibility in throughput, with values exceeding 0.1 g/min per nozzle. In contrast, unmodified PHBH was limited to lower throughput rates at a high process temperature of 190 °C, despite its lower melting temperature. This is attributed to a higher average polymer chain length and correspondingly a higher melt viscosity compared to P3HB. Nevertheless, PHBH and the PHBH+ATBC compounds led to a more stable meltblown process and more homogeneous fiber deposition due to the larger temperature difference between the process and melting temperatures.

It was also shown that the shape of the three-dimensional fabrics made from PHBH were completely compression-resilient, differing from P3HB and PHBV. However, the addition of plasticizers did not lead to any improvements without introducing inhomogeneities into the fiber deposition. However, the compounding of P3HB and PHBH with ATBC showed the potential to slightly reduce the processing temperatures of P3HB and PHBH, while significantly increasing the elongation and flexibility of PHBH. In contrast, the addition of ATBC resulted in a significant increase in fiber diameter for P3HB, which was generally not observed for PHBH.

As found in previous works, all PHAs showed semi-crystalline WAXS signals, proving that the meltblown process with high forces and strong chain orientation leads to highly crystalline structures without the influence of process parameter variations. The nonwovens made from pure PHBH showed the highest tenacity in both the machine direction (MD) and the cross direction (CD), while PHBV showed significantly lower mechanical strength. PHBV also exhibited the lowest elongation. The Young’s modulus was low for PHBV, while it was superior for P3HB with ELO. Furthermore, the use of PHBH led to higher tenacity at comparable fiber diameters and air permeability values. At the same time, fabric flexibility improved significantly, and elongation at maximum force increased in the MD. The PHBH compounds showed lower tenacity and air permeability but further increased elongation, fulfilling the hypothesis of this study. The brittleness of the P3HB nonwovens could not be significantly improved. Further investigations should now be carried out with other biological plasticizers. Furthermore, the mechanical recovery of the plant pots should be investigated over a longer period of stress. Due to the promising mechanical properties of the PHBH nonwovens, the meltblow processing of PHBH should now be further investigated. In the future, further applications for three-dimensional nonwovens made from PHAs can be investigated due to their interesting combination of air permeability and water retention properties, e.g., for use in the medical technology sector as a three-dimensional wound dressing.

## Figures and Tables

**Figure 1 polymers-17-00051-f001:**
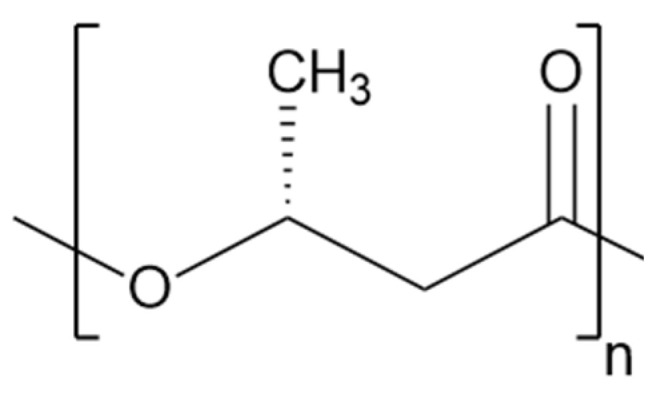
Chemical structure of P3HB.

**Figure 2 polymers-17-00051-f002:**
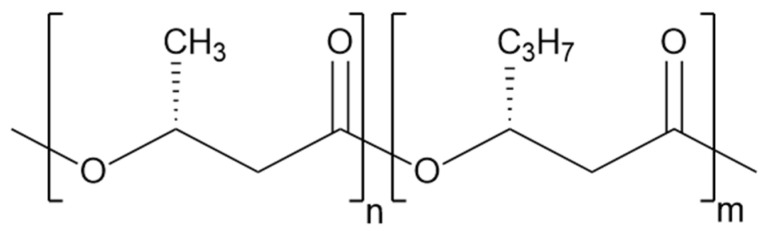
Chemical structure of PHBH.

**Figure 3 polymers-17-00051-f003:**
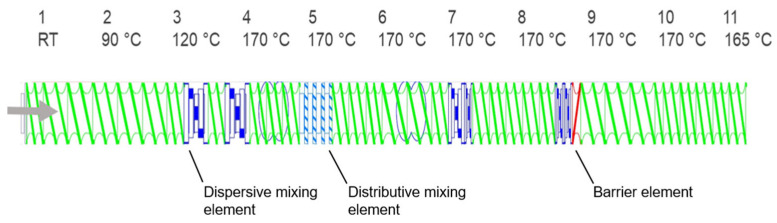
Screw configuration of the ZSK 26 twin-screw extruder and the associated temperature profile for the treatment (the arrow indicates the extrusion direction).

**Figure 4 polymers-17-00051-f004:**
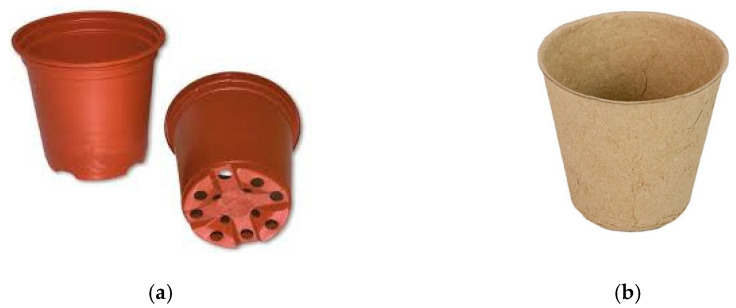
Commercial plant (seed) pots used as the counter-shape and reference samples. (**a**) Plastic plant pot (counter-shape); (**b**) cellulosic seed pot (biodegradable reference sample) [45].

**Figure 5 polymers-17-00051-f005:**
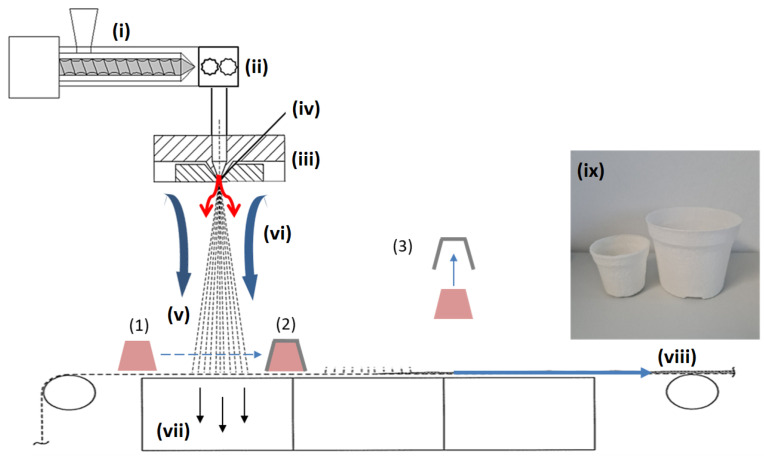
Schematic illustration of the three-dimensional nonwoven deposition on a counter-shape in the meltblow process. Process components: (**i**) extrusion system; (**ii**) gear pump; (**iii**) spin beam; (**iv**) nozzle (red arrows: primary (process) air stream); (**v**) polymer stream; (**vi**) secondary air (blue arrows); (**vii**) air suction; (**viii**) conveyor belt/winding; (**ix**) 3D demonstrator fabrics. Three-dimensional formation process: (**1**) counter-shape; (**1**) → (**2**) deposition of nonwoven layer on the counter-shape in the meltblow stream (**v**); (**3**) removal of 3D-fabric from counter.

**Figure 6 polymers-17-00051-f006:**
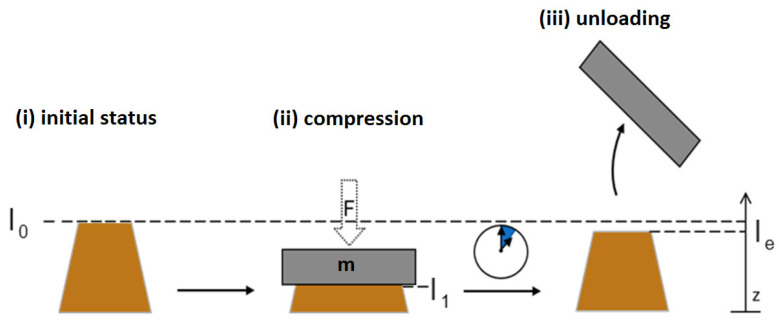
Schematic illustration of the qualitative determination of the flexibility and shape retention of the three-dimensional meltblown structures: (**i**) initial status (height *l*_0_) of the sample; (**ii**) sample compressed to height l_1_ by the weight of mass m; (**iii**) sample with height l_e_ after relaxation after unloading.

**Figure 7 polymers-17-00051-f007:**
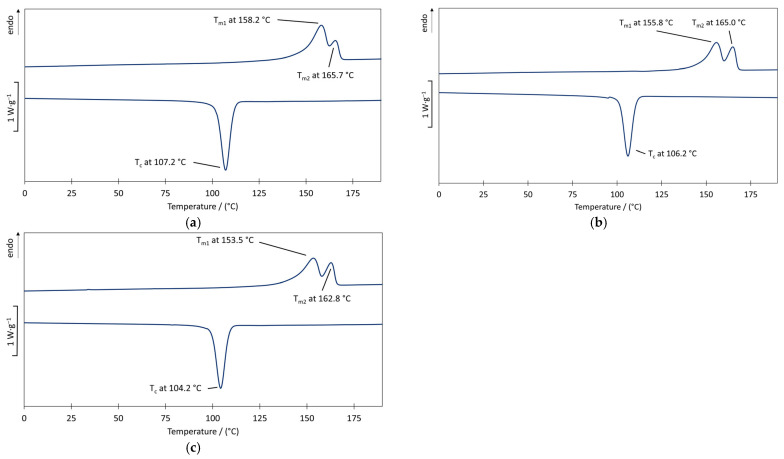
Illustration of the melting temperature Tm (second heating cycle) and crystallization temperature Tc (**a**) of commercially pure P3HB, (**b**) of P3HB with 10 wt.% ELO, and (**c**) of P3HB with 10 wt.% ATBC.

**Figure 8 polymers-17-00051-f008:**
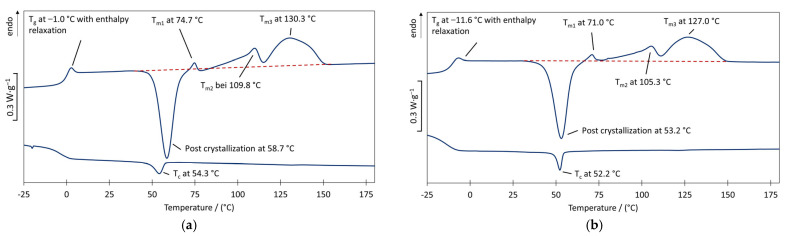
Illustration of the melting temperature *T_m_* (second heating cycle) and crystallization temperatures *T_c_* (**a**) of commercially pure PHBH and (**b**) PHBH with 10 wt.% ATBC (the dashed lines indicate the base line extension of the software for the evaluation of the melting peaks).

**Figure 9 polymers-17-00051-f009:**
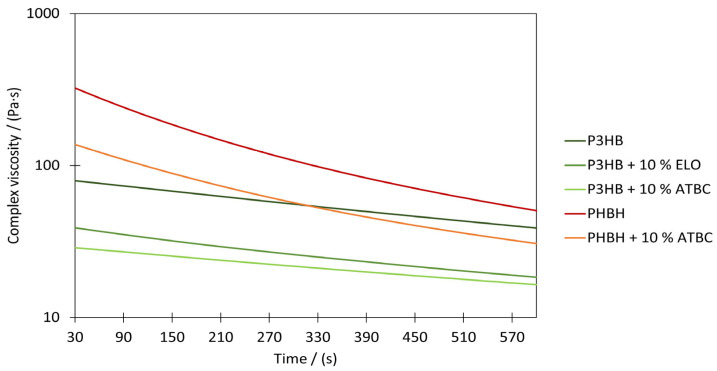
Rheological characterization using time sweep at a temperature of 180 °C, 5% strain, and a frequency of 1 rad s^−1^.

**Figure 10 polymers-17-00051-f010:**
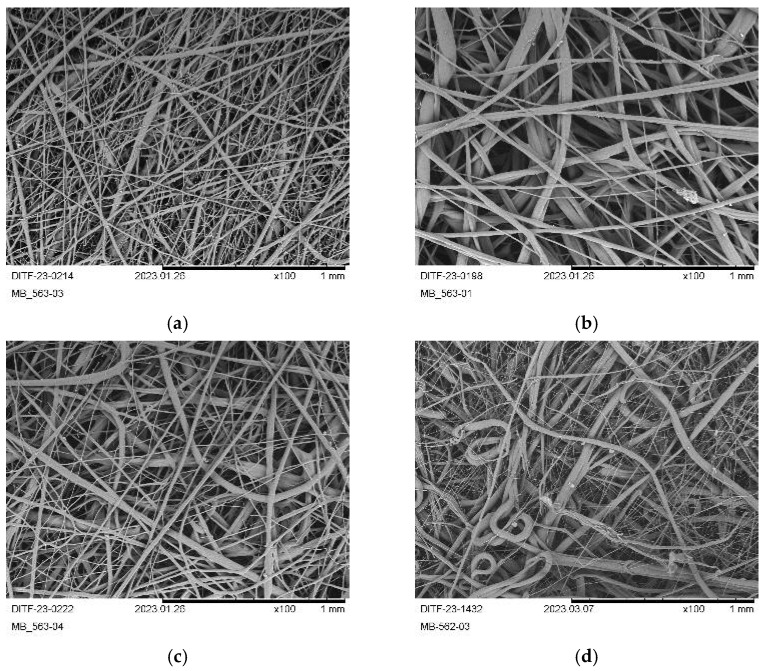
SEM images (100×) of the reference fabrics of P3HB and PHBV: (**a**) P3HB-Ref–01; (**b**) P3HB-Ref–02; (**c**) P3HB-Ref–03; (**d**) PHBV-Ref.

**Figure 11 polymers-17-00051-f011:**
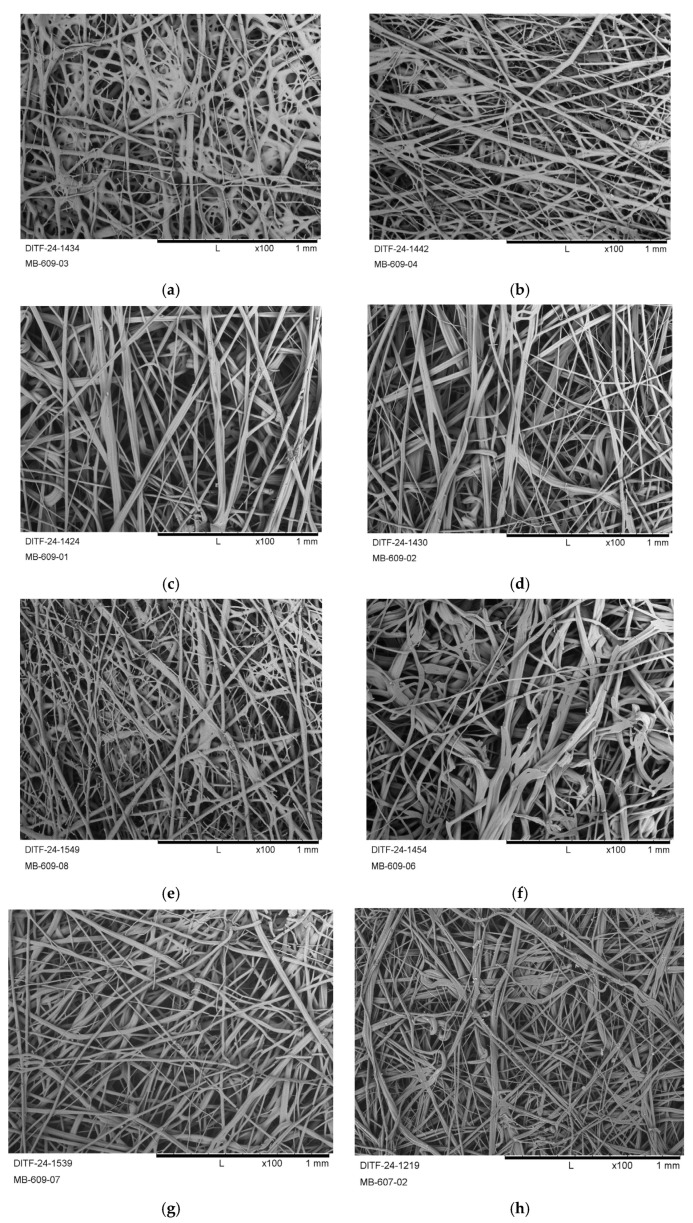
SEM pictures (100×) of the produced fabrics from P3HB+ELO, P3HB+ATBBC, PHBH and PHBH+ATBC: (**a**) P3HB+ELO–01; (**b**) P3HB+ELO–02; (**c**) P3HB+ELO–03; (**d**) P3HB+ELO–04; (**e**) P3HB+ATBC–01; (**f**) P3HB+ATBC–02; (**g**) P3HB+ATBC–03; (**h**) PHBH; (**i**) PHBH+ATBC–01; (**j**) PHBH+ATBC–02; (**k**) PHBH+ATBC–03.

**Figure 12 polymers-17-00051-f012:**
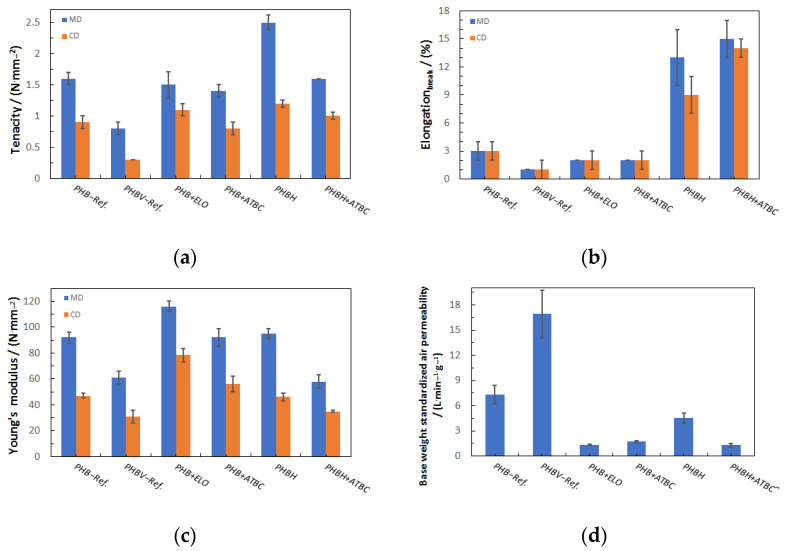
Comparison of the mechanical properties achieved with the different PHAs/compounds in MD (blue) and CD (orange): (**a**) tenacity, (**b**) elongation at break, (**c**) modulus, and (**d**) base-weight-standardized air permeability.

**Figure 13 polymers-17-00051-f013:**
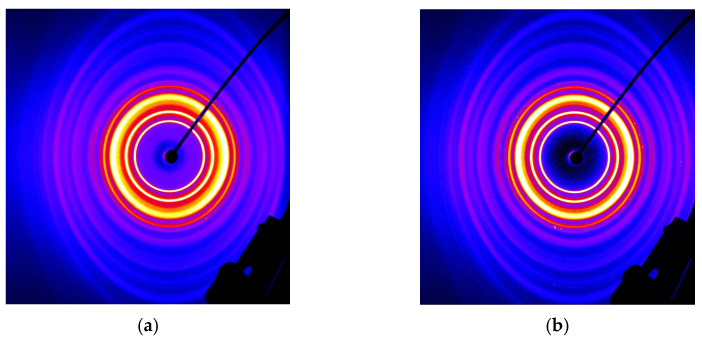
Exemplary X-ray scattering image of (**a**) a P3HB meltblown sample, (**b**) PHBV-ref., (**c**) and a PHBH meltblown sample and (**d**) corresponding X-ray diffraction patterns of different process settings: blue: P3HB; red: PHBV; green: PHBH.

**Figure 14 polymers-17-00051-f014:**
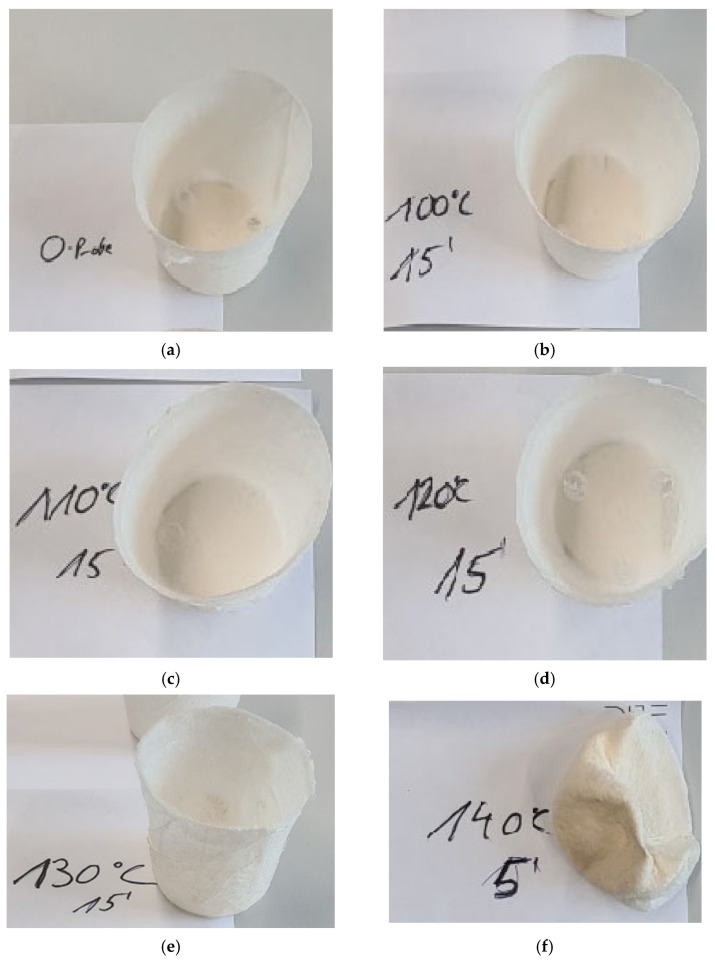
Results of the tests of 3D P3HB nonwoven samples regarding their thermal stability at different temperatures. (**a**) Reference—untreated; (**b**) 100 °C, 15 min; (**c**) 110 °C, 15 min; (**d**) 120 °C, 15 min; (**e**) 130 °C, 15 min; (**f**) 140 °C, 5 min; (**g**) 150 °C, 1 min.

**Table 1 polymers-17-00051-t001:** Results of thermogravimetric analysis.

Material	T_on_/°C	T_10%_/°C	T_end_/°C
P3HB	277.9	284.0	306.4
P3HB + 10 wt.% ELO	275.3	252.0	311.7
P3HB + 10 wt.% ATBC	276.2	244.7	306.8
PHBH	277.1	275.0	295.9
PHBH + 10 wt.% ATBC	276.6	269.0	297.3

**Table 2 polymers-17-00051-t002:** Rheological characteristics from the time sweeps (ω = 10 rad s^−1^, ε = 10%) and MFI data of the different PHAs.

Material	η_to_ (180 °C) /(Pa s)	G′_t0_ (180 °C) /Pa	G″_t0_ (180 °C) /Pa	MFI /(g 10 min^−1^) ^1^
P3HB	137	250	1354	10 [43]
PHBV	434	684	4287	10–25 [56]
PHBH	634	1150	6242	3 [44]
P3HB+ELO	37	30	373	-
P3HB+ATBC	38	33	380	-
PHBH+ATBC	245	252	2405	-

^1^ T = 180 °C, 2.16 kg.

**Table 3 polymers-17-00051-t003:** Estimation of the process temperature and characteristic shear rheological properties of the different PHAs at the process temperature.

Material	*T_proc_* /°C	*η_to_* (T_proc_) /(Pa·s)	*η*_300_*_s_* (T_proc_) /(Pa·s)	*G′_t_*_0_ (T_proc_) /Pa	*G″_t_*_0_ (T_proc_) /Pa
P3HB	180	137	67	250	1354
PHBV	200	227	49	199	2262
PHBH	190	325	52	151	3621
P3HB+ELO	176	48	36	47	476
P3HB+ATBC	176	45	37	41	445
PHBH+ATBC	190	62	18	17	623

**Table 4 polymers-17-00051-t004:** Process settings of the meltblow trials.

Trial-Nr.	T_melt_ /°C	T_air_ /°C	Throughput /(g ho^−1^ min^−1^)	Die-Pressure /bar	Air Volume Flow /(Nm^3^ h^−1^)	Limitation(s)
P3HB-Ref–01	180	175	0.077	31.0	220	–
P3HB-Ref–02	180	170	0.077	21.7	220	
P3HB-Ref–03	180	175	0.077	21.8	325	
PHBV-Ref	200	195	0.051	26.4	325	No stable process at lower throughput (degradation dominates) Limitation of max. throughput (die-pressure)
P3HB+ELO–01	176	170	0.039	7.1	220	Edges stick strongly to conveyor belt
P3HB+ELO–02	176	170	0.039	7.1	325	
P3HB+ELO–03	176	170	0.077	12.3	220	
P3HB+ELO–04	176	170	0.077	12.3	325	
P3HB+ATBC–01	176	170	0.039	23.5	220	–
P3HB+ATBC–02	176	170	0.077	25.0	220	The higher the throughput, the more inhomogeneous the deposition
P3HB+ATBC–03	176	170	0.077	25.0	325	
PHBH	190	185	0.032	35.0	325	Limitation of max. throughput (die pressure)
PHBH+ATBC–01	190	185	0.032	47.2	220	High pressure level → throughput limited
PHBH+ATBC–02	190	185	0.032	52.0	325	
PHBH+ATBC–03	190	178	0.039	46.4	240	

**Table 5 polymers-17-00051-t005:** Characteristics of the produced and reference nonwoven fabrics; bold: samples showing the best/superior properties.

Trial-Nr.	Base Weight		Thickness	Fiber Diameter		Air Permeability	Tenacity	Elongation ^2^	Modulus ^3^
	/(g m^−2^)	CV ^1^ /%	/µm	Median /µm	Mean /µm	/(L m^−2^ s^−1^)	MD/CD /(N mm^−2^)	MD/CD /%	MD/CD /(N mm^−2^)
P3HB–Ref–01	93	11	235 ± 19	4.6	7.0	680 ± 100	1.6 ± 0.1/0.9 ± 0.1	4 ± 1/5 ± 1	92 ± 4/47 ± 2
P3HB-Ref–02	94	14	346 ± 43	13.7	16.2	4640 ± 880	0.7 ± 0.1/0.5 ± 0.1	3 ± 1/3 ± 1	39 ± 7/28 ± 4
P3HB-Ref–03	99	16	299 ± 40	7.3	12.7	1550 ± 430	1.4 ± 0/0.7 ± 0.2	3 ± 0/3 ± 1	76 ± 5/38 ± 7
PHBV-Ref	120	8	566 ± 79	3.5	4.8	2030 ± 340	0.8 ± 0.1/0.3 ± 0	1 ± 0/1 ± 1	61 ± 5/31 ± 5
P3HB+ELO–01	220	5	456 ± 55	10.6	16.7	285 ± 14	1.5 ± 0.2/1.1 ± 0.1	2 ± 1/2 ± 1	**116 ± 4/78 ± 5**
P3HB-ELO–02	228	7	434 ± 34	10.5	12.8	230 ± 61	1.8 ± 0.1/1.0 ± 0.1	2 ± 0/2 ± 1	107 ± 5/69 ± 2
P3HB+ELO–03	223	12	562 ± 118	9.3	11.3	863 ± 156	1.1 ± 0.1/0.6 ± 0.1	3 ± 0/2 ± 1	53 ± 6/36 ± 4
P3HB+ELO–04	204	18	504 ± 132	7.3	9.8	537 ± 125	1.3 ± 0.1/1.0 ± 0.1	3 ± 0/2 ± 1	71 ± 6/54 ± 9
P3HB+ATBC–01	188	12	373 ± 52	8.8	12.0	390 ± 19	1.4 ± 0.1/0.8 ± 0.1	2 ± 0/2 ± 0	92 ± 7/56 ± 6
P3HB+ATBC–02	213	12	516 ± 102	14.2	15.9	765 ± 65	1.0 ± 0.1/0.6 ± 0.2	2 ± 0/2 ± 1	69 ± 12/36 ± 7
P3HB+ATBC–03	204	14	454 ± 106	7.4	10.0	543 ± 45	1.3 ± 0.1/0.8 ± 0.2	2 ± 1/2 ± 1	93 ± 17/47 ± 4
PHBH	128	11	390 ± 48	4.5	6.2	700 ± 69	**2.5 ± 0.1/1.2 ± 0.1**	5 ± 0/6 ± 0	100 ± 2/4 ± 1
PHBH+ATBC–01	140	12	335 ± 26	6.5	7.9	**185 ± 32**	1.5 ± 0.1/1.2 ± 0	10 ± 2/**22 ± 3**	60 ± 2/41 ± 2
PHBH+ATBC–02	131	11	343 ± 24	8.8	8.5	411 ± 101	1.6 ± 0/1.0 ± 0.1	**15 ± 2**/14 ± 1	58 ± 5/35 ± 1
PHBH+ATBC–03	120	22	376 ± 69	**3.6**	6.4	270 ± 40	2.2 ± 0.1/1.1 ± 0.1	6 ± 0/10 ± 2	85 ± 5/45 ± 2

^1^ Coefficient of variation (standard deviation divided by mean average). ^2^ Elongation at max. force. ^3^ Young’s modulus.

**Table 6 polymers-17-00051-t006:** Results of the heat shrinkage test (120 °C, 15 min) of the meltblown materials; the results comprise all variations of the respective PHA type/compounds.

Material	P3HB	P3HB+ECO	P3HB-ATBC	PHBV	PHBH	PHBH+ATBC
Shrinkage in MD/%	1.7 ± 0.6	1.9 ± 0.6	1.3 ± 0.5	1.6 ± 0.6	1.7 ± 0.8	1.5 ± 0.5
Shrinkage in CD/%	1 ± 1 ^1^					

^1^ No differences between all samples.

**Table 7 polymers-17-00051-t007:** Material characteristics of the three-dimensional PHA nonwoven demonstrators compared to the commercial reference plant pots.

Property	P3HB	PHBH	PHBV	Cellulose ^1^	Coconut ^1^
Area weight (g m^−2^)	151 ± 12	151 ± 22	148 ± 3	296 ± 18	476 ± 43
Wall thickness (µm)	646 ± 84	409 ± 48	662 ± 49	1597 ± 120	2689 ± 294
Tensile tenacity (N mm^−2^)	1.6 ± 0.1	2.2 ± 0.1	0.8 ± 0.2	1.6 ± 0.4	0.7 ± 0.2
Elongation (%)	4 ± 1	13 ± 3	2 ± 1	3 ± 1	2 ± 0
Youngs modulus (N mm^−2^)	75 ± 19	95 ± 4	49 ± 7	73 ± 13	49 ± 7
Air permeability (L m^−2^ s^−1^)	2010 ± 74	676 ± 75	1060 ± 56	27 ± 3	114 ± 2
Water retention (%)	100 ^2^	100 ^2^	100 ^2^	0 ^3^	0 ^3^

^1^ Commercial references. ^2^ For at least 50 h after filling. ^3^ Directly after filling.

**Table 8 polymers-17-00051-t008:** Results of the qualitative shape stability test (squeezing under load and unfolding); “0”: sample remains deformed after compression; “1”: full shape resiliency.

Material	PHB ^1^	PHBV	PHBH ^2^
Result	0.5 ^3^	0	1

^1^ Including PHB with additives (ELO/ATBC). ^2^ Including PHBH+ATBC. ^3^ Pods tear at the opening when loaded, but straighten up after the first time of reloading.

## Data Availability

Data available upon request due to privacy restrictions. The data presented in this study are available upon request from the corresponding author. The data are not publicly available due to running project issues.

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
