# Peer review of "Generation of Bio-Based, Shape- and Temperature-Stable Three-Dimensional Nonwoven Structures Using Different Polyhydroxyalkanoates"

_polymers, 2024, doi:10.3390/polym17010051_

Round 1

Reviewer 1 Report

Comments and Suggestions for Authors

The title should be concise. Is it possible to remove the long chemical names from Title.

The overall language is quite poor. It needs some major revisions. It seems it is translated from some other language to English. For example"However, the additivation of "

How the dozing rate and corresponding temperature was selected?

Use appropriate term for Nonwoven fabric weight

Why so much variation in fibre diameter? Please discuss in detail

The XRD peaks should be discussed in detail.

The conclusion should be concise and precise

Comments on the Quality of English Language

The overall language is quite poor. It needs some major revisions

Author Response

Dear Reviewer, Thank you very much for your time and effort to review our manuscript. Your feedback is very valuable to improve the quality of our work-. Please find subsequent the detailed answers on your comments. Comments and Suggestions for Authors Comment 1: The title should be concise. Is it possible to remove the long chemical names from Title. Answer 1: Thank you fort he comment. For our understanding it is required by the journal’s guidelines to avoid abbreviations, such as for chemical names, in the title. We’ll change tot he chemcial abbreviations P3HB, PHBH and PHBV, if it finds approval by the editor. Comment 2: The overall language is quite poor. It needs some major revisions. It seems it is translated from some other language to English. For example"However, the additivation of " Answer 2: Thank you for the comment. We appologize fort he lack in quality of the manuscript and revised the whole script briefly. Comment 3: How the dozing rate and corresponding temperature was selected? Answer 3: Thank you for this query! Based on tests with other PHA variants, a standardised temperature profile was defined in order to ensure the same external conditions for the subsequent material characterisations. This was defined in preliminary tests in order to obtain a homogeneous melt. The dosing rate of the polymer was determined according to the periphery for dosing the liquid additives, which can only produce a certain maximum volume flow depending on the viscosity. Comment 4: Use appropriate term for Nonwoven fabric weight Answer 4: Thank you! We changed the used term „base weight“ to „basis weight“ according tot he comment. Comment 5: Why so much variation in fibre diameter? Please discuss in detail Answer 5: We are not sure, wheather the quetions deals on the fiber diameter variation within one sample or on the fiber diameter variation between the samples. The fiber diameter variation within one samples is absed on the fiber diameter distribution and lies in common range of meltlbown nonwovens. Here, the fiber diameter distribution is generally broader distributed than in other processes, e.g. the melt spinning of yarns. The fiber diameter variation also agrees to previous research (Höhnemann, T., & Windschiegl, I. (2023). Influence of Rheological and Morphological Characteristics of Polyhydroxybutyrate on Its Meltblown Process Behavior. Materials, 16(19), 6525.“ ) on PHAs and is contributed to higher sensitivity to process parameter changes. However, we added a respective paragraph tot he discussion, l.555f: „The fiber diameter range, quantified by the mean and the median of the fiber diameter distribution, agree to the samples of previous work [41-43]. The average fiber diameters of the PHBH fabrics, as well as the fabrics of the modified polymers are slightly higher. However, the sensitivity of the fiber diameter to process parameter changes appears to be reduced. The fiber diameter variation of the samples (ratio of the media to the mean aveerage) also are in a common range of meltlbown nonwovens. In this contextm the meltlbown process generally generates broader fiber distributions distributed than in other processes, e.g. the melt spinning of yarns. This is reaseoned in the complex interactions between the polymer melt and the turbulent air flow [61,62]“ 61 Tan, D.H., Zhou, C., Ellison, C.J., Kumar, S., Macosko, C.W., Bates, F.S. Meltblown fibers: Influence of viscosity and elasticity on diameter distribution. J. Non-Newton. Fluid Mech. 2010 165(15-16), 892-900. 62 Ellison, C.J., Phatak, A., Giles, D.W., Macosko, C.W. Bates, F.S. . Melt blown nanofibers: Fiber diameter distributions and onset of fiber breakup. Polymer 2010 48(11), 3306-3316. Comment 6: The XRD peaks should be discussed in detail. Answer 6: Thank you for the remark! We revised the XRD part oft he mansucript: l.591ff „ Figure 13. Exemplary X-ray scattering image of (a) of a P3HB meltblown sample, (b) of PHBV-ref. and (cb) a PHBH meltblown sample and (d) corresponding X-ray diffraction patterns of different process settings; blue: P3HB; red: PHBV, green: PHBVH. As already obtained in our earlier previous work [4138] (compare sample “PHB–Ref” and “PHBV–Ref” in Figure 8b) all X-ray diffraction images showed semi-crystalline-pattern. signals. Several crystalline reflections occurred at 2θ ≈ 13.5° ((020)), 16.9° ((110)), 19.7° ((101)), 21.5° ((111)), 25.3° ((130)), 27.0° ((040)) and 30.0° ((002)) in accordance with the literature.[ 3,4138,6358] Differences in the process settings (throughput rates, air amounts) did not played no a role in the crystallization, which is characteristically starts delayed for PHAs. However, the physics of the meltblow process with high forces byfrom the air stream applying resulted in high(est) orientation, leading to resulting in highly crystallized structures. The crystallinity of all samples was determined The diffractograms were evaluated by fitting (pseudo-Voigt profiles) the diffractograms in the range of 2θ = 10°– 40°. In this way, tThe crystallinity for the P3HB samples was between was semi-quantitatively determined to achieve 71 % to 77 %, of OHBV between 75 % to 80 % for all P3HB samples and 60 % to 77 % for the PHBH samples. The plasticizers showed limited influence on the peak positions and crystallinity. In comparison to literature [3,4138,6358] the identified peaks indicate the predominance of orthorhombic α-crystals formation, with hexagonal β-form crystals (shoulder at 2θ ≈= 20°, (101)-lattice). This shoulder is indeed more pronounced for P3HB / PHBV and very diffuse for PHBH. Referring to Hufenus et al. [6460], β-crystals develop stress-introduced in amorphous P3HB regions between lamellar α-crystals. Further, the peak corresponding to the (111)-lattice is shifted slightly to lower -value. The presence of the shoulder at 2θ ≈ 20° ((101)-lattice) is typical for introduced chain orientation, such as uniaxially stretching [3,6559], stress-induced crystallization [10,6661] or distribution of different crystal sizes. Another reason can be random chain scission during degradation [6661–,6863] which is very commonhighly present in the processing of PHAs, forming crystals with different lamellar thicknesses and thus different melting kinetics. However, the results indicate that the copolymer prefers to form α-crystals rather than β-crystals. " Comment 7: The conclusion should be concise and precise Answer 7: We revised the conclusion according to the recommendation! Comments on the Quality of English Language Comment 8: The overall language is quite poor. It needs some major revisions Answer 8: Thank you for the comment. We appologize fort he lack in quality of the manuscript and revised the whole script briefly.

Reviewer 2 Report

Comments and Suggestions for Authors

Processing biobased and biodegradable polymers gained increasing interest during the last decade. This is especially valid for polymeric materials produced in a biotechnological process like PHAs. The present manuscript deals with the improvement of the processability and technological properties of this class of bio-polyesters by blending and compounding of P3HBH, a PHA copolymer that gained commercial availability only recently.   

Generally, PHAs are known to have limited materials properties, so every contribution to enhance the commercial applicability must be welcomed – and this manuscript is a good step forward in this direction.

The manuscript is very well structured, material and methods are described extensively, the results presented in Figures and Tables provide an extensive data base for further optimization of the materials properties bringing the materials closer to the market. 

As a minor weak point, I’m missing a more detailed investigation of the degradation during processing caused by thermal and/or mechanical stress, as the influence of molecular weight and its distribution look to me a little underestimated.  

As another minor correction the abbreviation 3HHx for 3-hydroxyhexanoate units should be used consistently throughout the paper (compare line 99, 366 and 395).

But overall, I recommend to accept this paper with minor corrections. 

Author Response

Dear Reviewer! Thank you very much for the postive feedback! Please find attached the detailed answers on your comments. Comment 1: As a minor weak point, I’m missing a more detailed investigation of the degradation during processing caused by thermal and/or mechanical stress, as the influence of molecular weight and its distribution look to me a little underestimated. Answer 1: Thank you for this note! The description of the influences on material behaviour in rheological characterisation has been extended. As this work is mainly concerned with the applicability and comparison of different material systems from the PHA family and the use of additives, ageing in its complexity is not explicitly dealt with here. Nevertheless, the state of the art has been expanded with a reference to current research work dealing with the ageing of PHAs. Further the influence of degradation on molar mass of PHAs was focussed in previous work: „Höhnemann, T., & Windschiegl, I. (2023). Influence of Rheological and Morphological Characteristics of Polyhydroxybutyrate on Its Meltblown Process Behavior. Materials, 16(19), 6525.“ Comment 2: As another minor correction the abbreviation 3HHx for 3-hydroxyhexanoate units should be used consistently throughout the paper (compare line 99, 366 and 395). Answer 2: Thank you for the note! The abbreviation for 3-hydroxyhexanoate was reviewed and adapted in the paper. Comment 3: But overall, I recommend to accept this paper with minor corrections. Answer 3: Thank you again very much!

Reviewer 3 Report

Comments and Suggestions for Authors

1. In Figure 5, the names of the various components should be marked to facilitate readers to understand. The title of Figure 5 should also be more detailed.

2. Equation (1) is not displayed in full.

3. In the title of Figure 6, the sub-figures should be described separately, and necessary information should be explained in the figure.

4. Line 472, there seems to be something wrong with the description of the sample number in Figure 10b. Please check and modify it.

5. “Plant pot demonstrators were exposed in to different temperatures starting at 100 °C in an oven for 15 minutes.”, please explain why the starting temperature was 100 °C? How was 15 minutes determined?

6. The conclusions should give the shortcomings and prospects of this study.

Author Response

Dear Reviewer,

Thank you very much for your time and effort to review our manuscript. Your constructive commetns helped significantly to improve the quality of our work-. Please find subsequent the detailed answers on your comments.

Comments and Suggestions for Authors

Comment 1: In Figure 5, the names of the various components should be marked to facilitate readers to understand. The title of Figure 5 should also be more detailed.

Answer 1: Thank you for the comment! We revised the Figure and marked the different components with numbers and added the their names respectively to the figure title in order for an easier understanding fort he readers.

Figure 5. Schematic illustration of the three-dimensional nonwoven deposition on a counter-shape in the meltblow process; Process components: (i) Extrusion system, (ii) gear pump, (iii) spinnbeam, (iv) nozzle (red: primary (process) air stream (v) polymer stream (vi) secondary air, (vii) air suction (viii) conveyor belt / winding (ix) 3D-demonstrator fabrics, 3D formation process: (1) counter-shape, (1) counter-shape, (1)→(2) deposition of nonwoven layer on counter in the meltblow-stream (v), (3) removal of 3D-fabric from counter.

Comment 2: Equation (1) is not displayed in full.

Answer 2: Thank you for the hint! Indeed, the equation was completely displayed in our initial submission. However, we inserted the equation again.

Comment 3: In the title of Figure 6, the sub-figures should be described separately, and necessary information should be explained in the figure.

Answer 3: Thank you for this comment! Figure 6 is only one figure and initially not divided into subfigures. As recommended, we added further necessary information in the figure and in the figure title in ordert to make the Figure more clear and to avoid confusion for readers.

Comment 4: Line 472, there seems to be something wrong with the description of the sample number

in Figure 10b. Please check and modify it.

Answer 4: Thank you for this hint! We revised the Figure title.

Figure 10. SEM images (×100) of the reference fabrics of P3HB and PHBV: (a) P3HB-Ref–01; (b)P3HB-Ref–02; (c) P3HB-Ref–03; (d) PHBV-Ref.

Comment 5: “Plant pot demonstrators were exposed in to different temperatures starting at 100 °C in an oven for 15 minutes.”, please explain why the starting temperature was 100 °C? How was 15 minutes determined?

Answer 5: Many thanks for the specific request question! The demosntrators were exposed to different temperatures starting at 100 °C for 15 minutes. Indeed, we tested also longer residence times with the first samples (30min, 1h), without without obtaining altered measurement results. Thus we decided on the time of 15minutes to conduct the experiments in reasonable time scale. This time of 15 min was further derived from different standards such as for measuring the heat shrinkage of yarns (DIN 53840 oder DIN EN 14621), fiber bundles (ASTM International - ASTM D2102-96) or fils / sheets (ISO 11501:1995 & GB/T 12027–2004)). The temperature was choosen in relation tot he caracteristic temperatures of the PHAs: above Tg (~0 °C) and below Tm (PHBH: 40 – 145 °C, P3HB / PHBV: ~175 °C)). This can be compared to PET fibers (with a Tg of around 70°C and a Tm of around 270°C), where shrinkage can be measured at 200°C.

Comment 6: The conclusions should give the shortcomings and prospects of this study.

Answer 6: Thank you! We revised the conclusion in oder to point out the shortcomings and prospects!

Round 2

Reviewer 1 Report

Comments and Suggestions for Authors

The authors have modified the manuscript as per comments